# Longitudinal associations between time perspective and life satisfaction across adulthood
Maria Wirth [1] ✉, Markus Wettstein [2] & Klaus Rothermund [1]

Time perspective is an important predictor of well-being. How time is represented, is itself subject to developmental change. A time perspective dominated by the future is increasingly replaced by one focused on the present and past as remaining lifetime decreases. These age-related changes supposedly are associated with higher subjective well-being. Previous studies yielded heterogeneous results. However, these studies mostly investigated one dimension of time perspective and did not include younger and/or middle-aged adults. Thus, we investigated how changes in four facets of time perspective (past-orientation, concreteness of future time, obsolescence, and attitudes towards finitude) were related to changes in life and domain-specific satisfaction and if these relations were moderated by age. We used 10-year longitudinal data from an age-diverse sample comprising 459 participants (30–80 years). Concreteness was most consistently related to satisfaction. Individuals with overall higher concreteness reported higher life satisfaction and higher life satisfaction was reported on measurement occasions with higher concreteness. An age moderation was only found for satisfaction with mental fitness. Among younger but not older adults, satisfaction with mental fitness was higher on measurement occasions with higher concreteness. Our study provides a deeper understanding of the relation between time perspective and well-being across adulthood.

The mental representation of past, present, and future time, known as time perspective[1] has been closely linked to well-being. How time is represented, is itself subject to developmental change. A time perspective dominated by the future is increasingly replaced by one focusing on the present and past as remaining lifetime decreases. These age-related changes in time perspective supposedly are associated with higher well-being[2,3]. Given that previous studies yielded heterogeneous results, our study aimed to enhance the understanding of the association between time perspective as a multi-dimensional construct and subjective well-being (SWB) from a lifespan perspective. We investigated the longitudinal relations between time perspective and one indicator of SWB, namely life satisfaction, in an age-diverse sample.

First, we review theoretical approaches to time perspective and its relation to SWB. Following, we discuss empirical work testing the proposed relations. We focus our review of empirical work on longitudinal studies that tested an age moderation of the relation between time perception and SWB (see refs. 4,5 for a review of cross-sectional studies). Generally, time perspective is conceptualized as a broad concept comprising attitudes and experiences relating to different facets of time (past, present, future,

individual and historical time and its changes). Although it has implications for and is assumed to be empirically related to other constructs (e.g., personality, motivation, affect), time perspective is a unique construct defined by its semantic relation to temporal aspects of an individual's experience.

Time perspective theory[6] conceptualizes time perspective as individual tendencies to emphasize a particular temporal frame (past, present, or future). Rigidly focusing on one time frame but neglecting the others should be related to lower well-being compared to a more balanced time perspective. Time perspective is conceptualized as relatively stable, thus, its relation to well-being could be relatively stable across adulthood. Socioemotional Selectivity Theory[3] (SST) focuses on future time perspective. SST assumes that if no or few future time limitations are perceived (typical for young adulthood), individuals prepare for that future by acquiring new skills and information. This should entail emotional costs and, thus, lower well-being. With a more limited future-time perspective (typical for late adulthood), individuals focus on the present, reduce negative experiences and emotions, and seek positive ones, thus, optimizing well-being[7]. Viewing the relation between changes in time perspective and well-being through the lens of SST is obstructed as the theory (1) only focuses on future time

[1]Department of Psychology, Friedrich Schiller University Jena, Am Steiger 3/1, Jena, Germany. [2]Department of Psychology, Humboldt University Berlin, Rudower Chaussee 18, Berlin, Germany. ✉e-mail: maria.wirth@uni-jena.de

perspective, (2) only on quantitative changes of future time perspective, and (3) links time perspective and well-being only indirectly via goal-pursuit and according actions.

A conception of time perspective that considers qualitative changes, directly links time perspective and well-being, and explicates changes in this relation across adulthood has been proposed by Brandtstädter[2,8,9]. Brandtstädter's conception of time perspective includes the future-related facets of affective valence (optimistic expectations for the future), concreteness (degree of temporal and content-related organization of the future), controllability (beliefs about how controllable and malleable the future will be), and openness (richness and appeal of the anticipated future). Additional facets are past-orientation (dominance of past vs. future-related thoughts), feelings of obsolescence (sense of falling out of time or not being able to keep up), and attitudes toward the finitude of life (attitudes towards approaching the end of life).

Perceiving future time as meaningful, full of opportunities, controllable, and deriving pleasure from future planning has been related to high SWB. A lack of such future prospects, at all ages, is related to lower well-being[9]. Decreases in future-related facets should be related to decreases in well-being. However, decreases in future-related facets are seen as more detrimental to well-being in young compared to late adulthood. In young adulthood, they are considered as expressions of hopelessness concerning the personal future, or as indicators of impulsive motivational tendencies that counteract long-term strivings. In late adulthood, changes in future-related facets can motivate a flexible adjustment of (life) goals and a selective deployment of resources. These mechanisms are assumed to dampen the distress caused by the age-related shortening of future lifetime.

The relation between past-orientation and well-being should also depend on age. A high past-orientation during young adulthood indicates a ruminating focus on personal failures and is related to low well-being. In late adulthood, past-orientation indicates a focus on past achievements and a sense of personal integrity[10]. Increases in past-orientation during late adulthood could help to maintain well-being even in the face of age-related losses[9]. Thus, increases in past-orientation should be detrimental to younger but not older adults' well-being. Concerning attitudes towards life's finitude, high levels and increases should be detrimental to well-being in young adulthood, indicating an excessive centering on death and dying. In late adulthood, developing a positive attitude towards life's finitude is an indication of accepting one's mortality and a more relaxed approach to death. Finally, high levels and increases in obsolescence are seen as detrimental to well-being, regardless of an individual's age. During young adulthood, feelings of obsolescence are seen as a loss of social embeddedness that can be elicited by (role) transitions (e.g., from adolescence to adulthood). Feelings of obsolescence might also arise due to an increased loss of abilities and might prompt the aging individual to let go of commitments that have become hard to maintain[11]. According to the framework proposed by Brandtstädter, changes in time perspective are not only related to time lived, i.e., chronological age, but also to subjective remaining life expectancy (SRLE). Chronological age and SRLE are assumed to have similar, albeit opposite, relations to time perspective. A person with low SRLE should perceive their future as less concrete and open and be more past-oriented than a person with high SRLE. Decreases in future-oriented facets are assumed to be more detrimental to well-being of individuals who have a high SRLE.

Taken together, time perspective theory[6] conceptualizes time perspective as a relatively stable, personality-like construct. SST and Brandtstädter conceptualize time perspective as malleable, cognitive-motivational construct that develops and changes as a function of experience over the lifespan. These age-related changes are supposedly beneficial for well-being.

Whether there is a relatively fixed relation between time perspective and well-being as can be deducted from the propositions of time perspective theory, is difficult to judge. To judge this, longitudinal studies are needed that assess changes in all postulated time perspectives, their relation to well-being, and consider age, ranging from young adulthood to old age, as moderator of this relation. Longitudinal studies within the SST framework

did not show that an increasingly limited future time perspective was related to higher well-being[12,13]. Only one study found an age moderation–the relation between an increasingly limited future time and lower well-being was less pronounced in old age[14].

Results of the only longitudinal study[15] investigating all facets of time perspective proposed by Brandtstädter also were not entirely consistent with theoretical ideas. They showed decreases in the future-related facets across a 4-year interval in a sample of middle-aged and older adults. This decrease was particularly pronounced in older adults. Past-orientation, obsolescence, and finitude showed time-related increases, which were particularly pronounced in older adults. Decreases in future-related facets and finitude but also increases in past-orientation and obsolescence were related to increases in depression. As predicted, the relation between changes in past-orientation and depression was moderated by age. In middle-aged participants, increases in past-orientation were related to increases in depression, whereas this relation was absent for older adults. Furthermore, SRLE was more closely related to changes in time perspectives than chronological age. Moderating effects of SRLE on the relation between changes in time perspective and well-being were not reported. Although this study showed the predicted age-dependent relation between changes in past-orientation and well-being, whether those were present for the other facets of time perspective remained unclear.

One potential explanation for previous theory-inconsistent findings is that qualitative changes in time perspective, which were only assessed in two studies, are more important for well-being than quantitative changes[16,17]. Additionally, all but one study used measures exclusively focusing on future time perspective. These measures were mostly unidimensional, thus, disregarding the multi-dimensional nature of time perspective[18,19]. Moreover, the studies did not include younger[12,15] and middle-aged adults[13,14,20]. Others only covered short observation periods ranging from 10 days[14] to less than one year[13]. Another problem in linking changes in time perspective to SWB is the discrepancy between the conceptualization and measurement of SWB[21]. According to the most prominent definition by Diener[22], SWB represents a person's cognitive and affective evaluation of their life. However, previous studies investigating the relation between time perspective and well-being assessed depression[12,15], psychological well-being[13], agitation and lonely dissatisfaction[20], and the frequency of positive and negative affect[14] as outcomes. These measures do not seem to represent general evaluations of an individual's life. All but one[15] previous longitudinal studies have also relied solely on chronological age as a moderator of the relation between changes in time perspective and well-being. Theoretical ideas about changes in time perspective across adulthood tie these changes to perceptions of remaining lifetime or life expectancy[3,9]. While chronological age and subjective remaining life expectancy (SRLE) are usually moderately inversely related[9], previous studies indicated that SRLE is more closely related to time perspective than chronological age[15]. Moreover, age may not be the best proxy for remaining lifetime, while SRLE quite accurately reflects objective nearness to death[23]. If remaining lifetime moderates associations between time perspective and well-being, SRLE might be a more adequate indicator for remaining lifetime than chronological age. It will therefore be important to investigate further whether the associations between time perspective and well-being are moderated by SRLE.

Taken together, only Brandtstädter's conception of time perspective directly links changes in time perspective to changes in well-being and explicates how these changes depend on chronological age and SRLE. Current research within this framework has partially confirmed the proposed relations and moderations but remains limited in terms of the sample age and observation period.

Given the importance of time perspective for well-being and previous theory-inconsistent findings, we investigated the relation between changes in time perspective and SWB in an age-diverse sample of the "Aging as Future" (AAF) study[24] spanning a 10-year interval. To take the multi-dimensionality of time perspective into account, we investigated changes in past-orientation, concreteness of future time, feelings of obsolescence, and attitudes towards finitude. We considered a possible moderation of the

relation between time perspectives and well-being by chronological age and SRLE. In contrast to previous studies and to be more consistent with the definition of SWB by Diener[22] as an evaluation of one's life, we focused on life satisfaction as SWB indicator. Capitalizing on the availability of domain-specific satisfaction in the AAF data set, we explored moderations of age and SRLE for the relation between time perspective and satisfaction in the domains of friends, health, mental fitness, and cognitive fitness. These domains have robustly proven to be of critical relevance for life satisfaction of individuals of different ages[17].

Following the reasoning of Brandtstädter and previous results[15], we expected that decreases in the assessed future-related facet, namely concreteness of future time, relate to decreases in life satisfaction. This relation between lower concreteness and lower life satisfaction should be more pronounced in younger than older adults. Increases in past-orientation should be related to lower and decreasing life satisfaction in younger but less in older adults. Similarly, a stronger focus on life's finitude should be related to decreases in life satisfaction in younger but not older adults. Increases in obsolescence were expected to be related to lower life satisfaction at all ages. We also tested moderating effects of SRLE on the relation between time perspectives and life satisfaction. We expected that an older chronological age corresponds to lower SRLE and that similar moderation effects of an older age vs. a shorter SRLE would emerge. Additionally, we explored if our results depended on satisfaction domain (friends, health, mental fitness, and cognitive fitness).

## Methods
### Sample and procedure
We used data from the German subsample of the AAF longitudinal study[24]. The initial sample at T1 comprised $N = 768$ participants aged 30–80 years ($M_{age} = 55.27$, $SD_{age} = 14.85$, 49.5% female). The AAF used stratified random sampling (balanced design) and the sample was stratified by age cohort ($1929$–$1938 \times 1939$–$1948 \times 1949$–$1958 \times 1959$–$1968 \times 1969$–$1978$) and sex (male $\times$ female). For the present study, we only included participants who took part in all three measurement occasions (2009, 2014, and 2019). This sample included 459 participants aged 30–80 years at T1 ($M_{age} = 54.22$, $SD_{age} = 13.80$, 51% female). 120 individuals only participated in T1, 71 in T2, and 112 only in T3. 134 individuals participated only in T1 and T2, 55 in T1 and T3, and 164 in T2 and T3. We observed no systematic differences between AAF participants who participated only in T1 compared to those who took part more than once[25]. Sample characteristics for each cohort can be found in Supplementary Table 1.

Participants were recruited from two German cities, Jena and Erlangen, using information obtained from local registry offices. After providing written informed consent, questionnaires were sent via mail to participants' homes. Upon returning the completed questionnaire, participants received a gift card of approximately $20. Research procedures were approved by the Institutional Review Board at Friedrich-Schiller-University Jena (FSV 18/36). Details on ethics approval, sampling, and assessment procedure can be found in Lang and colleagues[24].

The study was not pre-registered.

### Measures
The measures used in the current study were part of a larger questionnaire[24]. The questionnaire included a wide range of variables, for this study, we investigated the following:

**Time perspective**. Time perspective was assessed with a short version of the Time Perspective Questionnaire[9]. The AAF data set includes three items each for the subscales concreteness of future time (e.g., "I have clear future-related goals that I pursue."), past-orientation (e.g., "I think more often about my past life than about my future."), feelings of obsolescence (e.g., "I increasingly have the feeling that I have lost touch with modern times") and attitude towards finitude (e.g., "I face the end of life with serenity."). Responses were given on a 5-point scale ranging from 0 ("strongly disagree") to 4 ("strongly agree"). Scale scores were derived by

summing responses for each scale for each measurement point. Internal consistencies for the scales ranged between 0.60–0.74 at T1, 0.67–0.74 at T2, and 0.64–0.77 at T3. Intraclass correlations (ICC) were 0.497 (obsolescence), 0.661 (past orientation), 0.535 (concreteness), and 0.610 (finitude); thus, 50% to 66% of the variation in time perspectives is due to between-person differences, whereas 34% to 50% of the variation is due to within-person variability. The ICCs show that all outcomes reveal a substantial proportion of intraindividual variability, justifying the use of longitudinal multilevel regression models.

**Chronological age**. Chronological age (in years) at T1 was included as continuous variable. The range of age at T1 was 30 to 80 years.

**Life satisfaction and domain satisfaction**. We assessed life satisfaction with a scale used in previous studies[26,27]. Participants indicated their domain-specific satisfaction concerning friends, leisure, personality, finances, work, physical fitness, mental fitness, appearance, and health. Responses were given on a 5-point scale ranging from 0 ("very unsatisfied") to 4 ("very satisfied"). Domains that were not assessed in the same way across measurement occasions were not included in the overall score for life satisfaction. Specifically, satisfaction with family and satisfaction with partnership were assessed as two items at the first measurement occasion, but as one item on subsequent measurement occasions. For overall life satisfaction, responses across all domains were aggregated and averaged at each measurement occasion. Internal consistency of the life satisfaction scale from T1 to T3 was $\alpha = 0.79$, $\alpha = 0.79$, and $\alpha = 0.86$. ICC for life satisfaction was 0.518, indicating that slightly more than half of the variation in life satisfaction was due to interindividual differences and the remaining proportion was due to within-person variability.

**Subjective remaining life expectancy (SRLE)**. SRLE for T1 was calculated as the difference between participants' age and their response to the statement "I believe that I am probably going to be ___ years old."[15]. Higher values represent longer expected time left to live.

**Covariates**. We controlled in our analysis for sex, primary education, monthly income (in Euro; for the analyses, we rescaled income in units of thousands, so that a difference in income by one unit corresponds to a difference of 1000 Euro), and subjective health reported at T1. Subjective health was assessed by a single item, "How would you rate your current health?" with a response scale from 0 (very poor) to 4 (very good). For the analyses, primary education was dummy-coded, using Volks-/Hauptschulabschluss (<10 years of formal education) as reference category. For those individuals who reported "other educational degree" ($N = 6$), the education score was set to missing. Two individuals reported an income that was more than 3 SD above the sample mean. We considered their income scores as outliers and set them to missing.

### Statistics and reproducibility
Due to missing values on certain study variables (e.g., income) and exclusion of outliers, the final sample size was reduced to $N = 420$. Longitudinal multilevel regression models were computed using SAS PROC MIXED[28,29]. Time in the study (in years) was used as time metric. The time perspective scales were considered as time-varying predictors. Following common practice[30,31], between-person and within-person effects were differentiated. Between-person effects describe to what extent those with overall higher scores on the respective time perspective scale (with the score averaged across all available measurement occasions per person) have higher or lower life satisfaction scores than those with overall lower scores on the time perspective scale. The person-specific mean of each participant on the time perspective scales across all available measurement occasions corresponds to the between-person effects of these predictors. We grand-mean-centered the between-person predictors at the overall sample means.

Within-person effects indicate whether scoring higher or lower than usual on the time perspective scales on a given measurement occasion is associated with scoring higher or lower than usual on life satisfaction at that specific measurement occasion. The deviation from an individual's person-specific mean at a given measurement occasion corresponds to the within-person predictor effect, which varies across time. The resulting model equations, including covariates and moderating effects of (1) age vs. (2) SRLE can be found in Supplementary Note 1. Individual–specific random effects around the intercept and the slope were specified, to account for interindividual differences in baseline scores and in changes over time. We also included time–specific residuals.

In the longitudinal multilevel regression models, we controlled for covariates (sex, education, income, self-rated health), specified as time-invariant predictors. We included age at T1 as well as SRLE at T1 and specified interaction terms of age and SRLE with the time perspective scales to identify potential moderating effects of age/SRLE on associations between time perspective scales and life satisfaction. As associations of age or SRLE with life satisfaction and the moderation effects of age or SRLE might be nonlinear, we included quadratic age and SRLE terms as well, which were trimmed from the final models whenever they did not reach statistical significance. Given the high correlation between age and SRLE, $r(446) = -0.89$, we decided to run separate models.

### Reporting summary
Further information on research design is available in the Nature Portfolio Reporting Summary linked to this article.

## Results
### Descriptive results
Supplementary Table 2 shows means and Table 1 correlations of the relevant study variables. Time perspectives and life satisfaction had substantial temporal stability of moderate and comparable size. Facets of time perspective had small to medium-sized intercorrelations, except attitudes towards finitude, which was not significantly related with most other time perspective facets. Concreteness, past-orientation, and obsolescence showed significant, medium correlations with life satisfaction. Attitudes toward finitude were not significantly related to life satisfaction.

Table 2 and Supplementary Fig. 1 summarize the model-implied trajectories of the time perspectives by age. Initial scores in obsolescence were not related to age, but the mean-level within-person increase over time was steeper in older individuals ($\gamma_{age*time} = 0.004$, df = 904, CI [0.002, 0.005], $p < 0.0001$). Chronologically older adults had higher past-orientation scores at T1, ($\gamma_{age} = 0.023$, df = 457, CI [0.007, 0.039], $p = 0.005$) and exhibited a steeper increase in past-orientation over time, ($\gamma_{age*time} = 0.003$, df = 904, CI [0.001, 0.005] $p = 0.003$). Chronologically older individuals reported lower concreteness at T1 ($\gamma_{age} = -0.047$, df = 457, CI [−0.062, −0.032], $p < 0.0001$). There was a mean-level decline in concreteness over time ($\gamma_{time} = -0.047$, df = 904, CI [−0.071, −0.023], $p < 0.0001$) which was not significantly moderated by chronological age. Chronologically older adults had higher finitude scores at T1 ($\gamma_{age} = 0.023$, df = 457, CI [0.006, 0.041], $p = 0.010$). Neither change in finitude over time nor the moderating effect of age were significant. Concerning effect sizes, the relative reduction in residual variance[32] ($R^2$) that was obtained by specifying age, time in study, and their interaction as predictors of time perspectives was small for finitude and concreteness ($R^2 = 0.05$ and $R^2 = 0.08$), but higher for past orientation and obsolescence ($R^2 = 0.13$ and $R^2 = 0.23$).

### Time perspectives with life satisfaction: chronological age moderation
We computed a longitudinal multilevel regression model, specifying the time perspective scales as between-person and within-person predictors of life satisfaction trajectories (Table 3). Covariates were controlled for and interactions of the time perspective scales with age were specified. The between-person effects of concreteness ($\gamma = 0.072$, df = 404, CI [0.047, 0.085], $p < 0.001$) and obsolescence ($\gamma = -0.039$, df = 404, CI [−0.066,

−0.022]. $p < 0.0001$) were significant. Individuals with overall higher concreteness scores and those with overall lower obsolescence scores reported greater life satisfaction (Supplementary Fig. 2). A concreteness higher by 1 $SD$ was associated with a life satisfaction score that was higher by ¼ $SD$. An obsolescence score that was higher by 1 $SD$ was associated with a life satisfaction score that was lower by 13% of 1 $SD$. On the within-person level, only concreteness ($\gamma = 0.033$, df = 810, CI [0.018, 0.049], $p < 0.0001$) was significantly related to life satisfaction. On measurement occasions when individuals had higher concreteness scores than usual, they also reported higher life satisfaction (Supplementary Fig. 3). This effect was small, indicating that a within-person concreteness score that is higher by 1 $SD$ was associated with a life satisfaction score that was higher by 7% of 1 $SD$. None of the interaction terms of time perspective scales with chronological age (and with quadratic chronological age; these interactions were removed from the final model) were significant. All predictors together accounted for 21% of the variance in life satisfaction.

### Associations of time perspectives with life satisfaction: SRLE moderation
When replacing chronological age with SRLE (Table 3), the significant between-person associations of concreteness ($\gamma = 0.066$, df = 404, CI [0.047, 0.085], $p < 0.0001$) and obsolescence ($\gamma = -0.044$, df = 404, CI [−0.066, −0.022], $p < 0.0001$) with life satisfaction were replicated as well as the significant within-person association of concreteness with life satisfaction ($\gamma = 0.033$, df = 810, CI [0.018, 0.049], $p < 0.0001$). Coefficients were in the same effect size range as in the model with chronological age. The within-person association between obsolescence and life satisfaction was statistically significant ($\gamma = -0.017$, df = 810, CI [−0.034, −0.000], $p = 0.038$), indicating that individuals reported higher life satisfaction on measurement occasions when their obsolescence scores were lower. This effect was small, as a within-person obsolescence score higher by 1 $SD$ was associated with a life satisfaction score that was lower by 5% of 1 $SD$. None of the associations of time perspectives with life satisfaction were significantly moderated by SRLE (or by a quadratic SRLE component; these interactions were removed from the final model). All predictors together accounted for 21% of the variance in life satisfaction.

### Additional analyses
Multilevel regression models were recomputed by replacing general life satisfaction with satisfaction in the domains of friends (ICC = 0.38), health (ICC = 0.40), physical fitness (ICC = 0.44), and mental fitness (ICC = 0.36). We focused on potential moderations of the relations by age or SRLE. We did not find evidence that age or SRLE moderated the effect of time perspectives on satisfaction with friends (Supplementary Table 3) or health (Supplementary Table 4). For satisfaction with physical fitness (Supplementary Table 5), there was no evidence of an age moderation but a significant SRLE moderation effect ($\gamma_{finitude*srle} = 0.003$, df = 404, CI [0.001, 0.005], $p = 0.015$). As can be seen in Fig. 1, higher finitude scores were associated with greater satisfaction with physical fitness among those with higher SRLE, but the association was slightly negative among those with lower SRLE.

For satisfaction with mental fitness (Supplementary Table 6), there was no evidence of a moderating effect of SRLE but two significant age moderation effects. As can be seen in Fig. 2a, overall higher past orientation was associated with lower satisfaction with mental fitness in younger, but not in chronologically older adults. As shown in Fig. 2b, among younger but not older adults, satisfaction with mental fitness was higher on measurement occasions with higher concreteness.

## Discussion
We investigated the longitudinal relation between time perspective, life and domain-specific satisfaction, and whether this relation was moderated by chronological age or SRLE. Changes in time perspective were mostly consistent with previous theoretical ideas and empirical findings. Concerning the relation between time perspective and well-being, between-person

**Table 1 | Correlations among time perspectives, life satisfaction, age, and SRLE**

| | 1 | 2 | 3 | 4 | 5 | 6 | 7 | 8 | 9 | 10 | 11 | 12 | 13 | 14 | 15 | 16 |
|---|---|---|---|---|---|---|---|---|---|---|---|---|---|---|---|---|
| 1. Age$_{T1}$ | - | | | | | | | | | | | | | | | |
| 2. CON$_{T1}$ | −0.28 [−0.37, −0.20] $p < 0.001$ | - | | | | | | | | | | | | | | |
| 3. CON$_{T2}$ | −0.25 [−0.33, −0.16] $p < 0.001$ | 0.60 [0.54, 0.66] $p < 0.001$ | - | | | | | | | | | | | | | |
| 4. CON$_{T3}$ | −0.26 [−0.34, −0.17] $p < 0.001$ | 0.48 [0.41, 0.55] $p < 0.001$ | 0.57 [0.50, 0.63] $p < 0.001$ | - | | | | | | | | | | | | |
| 5. PAST$_{T1}$ | 0.15 [0.06, 0.24] $p = 0.086$ | −0.20 [−0.29, −0.11] $p = 0.001$ | −0.19 [−0.28, −0.10] $p = 0.003$ | −0.23 [−0.31, −0.14] $p < 0.001$ | - | | | | | | | | | | | |
| 6. PAST$_{T2}$ | 0.15 [0.06, 0.24] $p = 0.065$ | −0.21 [−0.29, −0.12] $p < 0.001$ | −0.22 [−0.30, −0.13] $p < 0.001$ | −0.27 [−0.36, −0.19] $p < 0.001$ | 0.60 [0.53, 0.65] $p < 0.001$ | - | | | | | | | | | | |
| 7. PAST$_{T3}$ | 0.27 [0.18, 0.35] $p < 0.001$ | −0.21 [−0.30, −0.12] $p < 0.001$ | −0.22 [−0.30, −0.13] $p < 0.001$ | −0.24 [−0.33, −0.15] $p < 0.001$ | 0.53 [0.46, 0.59] $p < 0.001$ | 0.58 [0.51, 0.64] $p < 0.001$ | - | | | | | | | | | |
| 8. OBS$_{T1}$ | 0.05 [−0.04, 0.14] $p > 0.999$ | −0.30 [−0.38, −0.22] $p < 0.001$ | −0.26 [−0.34, −0.17] $p < 0.001$ | −0.17 [−0.26, −0.08] $p = 0.021$ | 0.35 [0.27, 0.43] $p < 0.001$ | 0.26 [0.17, 0.34] $p < 0.001$ | 0.26 [0.18, 0.35] $p < 0.001$ | - | | | | | | | | |
| 9. OBS$_{T2}$ | 0.04 [−0.05, 0.14] $p > 0.999$ | −0.26 [−0.34, −0.17] $p < 0.001$ | −0.27 [−0.35, −0.17] $p < 0.001$ | −0.25 [−0.33, −0.16] $p < 0.001$ | 0.36 [0.28, 0.44] $p < 0.001$ | 0.44 [0.36, 0.51] $p < 0.001$ | 0.35 [0.27, 0.43] $p < 0.001$ | 0.61 [0.55, 0.66] $p < 0.001$ | - | | | | | | | |
| 10. OBS$_{T3}$ | 0.22 [0.14, 0.31] $p < 0.001$ | −0.26 [−0.34, −0.17] $p < 0.001$ | −0.25 [−0.33, −0.16] $p < 0.001$ | −0.20 [−0.29, −0.11] $p = 0.001$ | 0.35 [0.26, 0.42] $p < 0.001$ | 0.37 [0.29, 0.45] $p < 0.001$ | 0.48 [0.41, 0.55] $p < 0.001$ | 0.47 [0.39, 0.54] $p < 0.001$ | 0.51 [0.44, 0.58] $p < 0.001$ | - | | | | | | |
| 11. FIN$_{T1}$ | 0.13 [0.04, 0.22] $p = 0.266$ | 0.16 [0.07, 0.25] $p = 0.039$ | 0.12 [0.03, 0.21] $p = 0.388$ | 0.06 [−0.03, 0.15] $p > 0.999$ | −0.03 [−0.12, 0.07] $p > 0.999$ | −0.04 [−0.13, 0.06] $p > 0.999$ | −0.01 [−0.09, 0.09] $p > 0.999$ | −0.13 [−0.22, −0.04] $p = 0.207$ | −0.06 [−0.16, 0.03] $p > 0.999$ | −0.03 [−0.12, 0.07] $p > 0.999$ | - | | | | | |

## Table 1 (continued) | Correlations among time perspectives, life satisfaction, age, and SRLE

| | 1 | 2 | 3 | 4 | 5 | 6 | 7 | 8 | 9 | 10 | 11 | 12 | 13 | 14 | 15 | 16 |
|---|---|---|---|---|---|---|---|---|---|---|---|---|---|---|---|---|
| 12. FIN$_{T2}$ | 0.11 | 0.10 | 0.12 | 0.05 | −0.09 | −0.07 | −0.11 | −0.12 | −0.11 | −0.03 | 0.64 | - | | | | |
| | [0.02, 0.20] | [0.01, 0.19] | [0.02, 0.21] | [−0.04, 0.14] | [−0.18, 0.00] | [−0.16, 0.03] | [−0.20, −0.02] | [−0.21, −0.03] | [−0.20, −0.02] | [−0.13, 0.06] | [0.58, 0.69] | | | | | |
| | $p = 0.797$ | $p = 0.518$ | $p = 0.518$ | $p > 0.999$ | $p > 0.999$ | $p > 0.999$ | $p = 0.607$ | $p = 0.361$ | $p = 0.743$ | $p > 0.999$ | $p < 0.001$ | | | | | |
| 13. FIN$_{T3}$ | 0.17 | 0.07 | 0.05 | 0.06 | −0.02 | −0.01 | −0.01 | −0.13 | −0.11 | −0.07 | 0.59 | 0.60 | - | | | |
| | [0.08, 0.26] | [−0.03, 0.16] | [−0.04, 0.14] | [−0.03, 0.15] | [−0.11, 0.08] | [−0.09, 0.09] | [−0.11, 0.08] | [−0.22, −0.04] | [−0.20, −0.01] | [−0.16, 0.03] | [0.53, 0.65] | [0.54, 0.65] | | | | |
| | $p = 0.015$ | $p > 0.999$ | $p > 0.999$ | $p > 0.999$ | $p > 0.999$ | $p > 0.999$ | $p > 0.999$ | $p = 0.207$ | $p = 0.839$ | $p > 0.999$ | $p < 0.001$ | $p < 0.001$ | | | | |
| 14. SAT$_{T1}$ | 0.16 | 0.30 | 0.27 | 0.24 | −0.23 | −0.16 | −0.13 | −0.36 | −0.28 | −0.20 | 0.13 | 0.10 | 0.13 | - | | |
| | [0.07, 0.25] | [0.21, 0.38] | [0.18, 0.35] | [0.15, 0.32] | [−0.32, −0.14] | [−0.25, −0.07] | [−0.22, −0.03] | [−0.44, −0.28] | [−0.37, −0.20] | [−0.29, −0.11] | [0.04, 0.22] | [0.00, 0.19] | [0.04, 0.22] | | | |
| | $p = 0.031$ | $p < 0.001$ | $p < 0.001$ | $p < 0.001$ | $p < 0.001$ | $p = 0.001$ | $p = 0.333$ | $p < 0.001$ | $p < 0.001$ | $p < 0.001$ | $p = 0.270$ | $p > 0.999$ | $p = 0.292$ | | | |
| 15. SAT$_{T2}$ | 0.12 | 0.22 | 0.31 | 0.26 | −0.19 | −0.18 | −0.11 | −0.30 | −0.32 | −0.25 | 0.06 | 0.09 | 0.09 | 0.64 | - | |
| | [0.03, 0.21] | [0.13, 0.30] | [0.23, 0.39] | [0.17, 0.35] | [−0.28, −0.10] | [−0.26, −0.08] | [−0.20, −0.02] | [−0.39, −0.22] | [−0.40, −0.24] | [−0.34, −0.16] | [−0.04, 0.15] | [0.00, 0.18] | [−0.01, 0.18] | [0.58, 0.69] | | |
| | $p = 0.456$ | $p < 0.001$ | $p < 0.001$ | $p < 0.001$ | $p = 0.004$ | $p = 0.011$ | $p = 0.682$ | $p < 0.001$ | $p < 0.001$ | $p < 0.001$ | $p > 0.999$ | $p > 0.999$ | $p > 0.999$ | $p < 0.001$ | | |
| 16. SAT$_{T3}$ | 0.02 | 0.28 | 0.29 | 0.35 | −0.21 | −0.21 | −0.21 | −0.28 | −0.26 | −0.28 | 0.02 | 0.06 | 0.02 | 0.46 | 0.53 | - |
| | [−0.07, 0.11] | [0.20, 0.37] | [0.20, 0.37] | [.27, 0.43] | [−0.29, −0.12] | [−0.30, −0.12] | [−0.30, −0.12] | [−0.36, −0.19] | [−0.35, −0.18] | [−0.37, −0.20] | [−0.07, 0.11] | [−0.04, 0.15] | [−0.07, 0.11] | [0.38, 0.53] | [0.46, 0.59] | |
| | $p > 0.999$ | $p < 0.001$ | $p < 0.001$ | $p < 0.001$ | $p < 0.001$ | $p < 0.001$ | $p < 0.001$ | $p < 0.001$ | $p < 0.001$ | $p < 0.001$ | $p > 0.999$ | $p > 0.999$ | $p > 0.999$ | $p < 0.001$ | $p < 0.001$ | |
| 17. SRLE$_{T1}$ | −0.89 | 0.29 | 0.27 | 0.29 | −0.20 | −0.17 | −0.28 | −0.12 | −0.09 | −0.27 | −0.11 | −0.09 | −0.14 | −0.03 | −0.02 | 0.06 |
| | [−0.91, −0.87] | [0.20, 0.37] | [0.18, 0.35] | [0.21, 0.38] | [−0.29, −0.11] | [−0.26, −0.08] | [−0.36, −0.19] | [−0.21, −0.03] | [−0.18, 0.00] | [−0.35, −0.18] | [−0.20, 0.01] | [−0.18, 0.00] | [−0.23, −0.04] | [−0.12, 0.06] | [−0.11, 0.08] | [−0.04, 0.15] |
| | $p < 0.001$ | $p < 0.001$ | $p < 0.001$ | $p < 0.001$ | $p = 0.001$ | $p = 0.016$ | $p < 0.001$ | $p = 0.384$ | $p > 0.999$ | $p < 0.001$ | $p = 0.871$ | $p > 0.999$ | $p = 0.203$ | $p > 0.999$ | $p > 0.999$ | $p > 0.999$ |

*p values were adjusted for multiple comparisons using the method proposed by Holm (1979).*
*CON concreteness of future time perspective, PAST orientation towards the past, OBS feelings of obsolescence, FIN attitudes toward the finitude of life, SAT satisfaction with life, SRLE subjective remaining life expectancy.*

**Table 2 | Growth models of time perspectives**

| | Obsolescence | | | | Past-Orientation | | | |
|---|---|---|---|---|---|---|---|---|
| | Estimate | SE | p | 95% CI | Estimate | SE | p | 95% CI |
| **Fixed effects** | | | | | | | | |
| Intercept | 2.857 | 0.096 | <0.0001 | 2.67; 30.045 | 4.805 | 0.113 | <0.0001 | 4.583; 50.026 |
| Age | 0.002 | 0.007 | 0.766 | −0.012; 0.016 | 0.023 | 0.008 | 0.005 | 0.007; 0.039 |
| Time | 0.095 | 0.012 | <0.0001 | 0.072; 0.119 | 0.041 | 0.013 | 0.002 | 0.016; 0.067 |
| Age*Time | 0.004 | 0.001 | <0.0001 | 0.002; 0.005 | 0.003 | 0.001 | 0.003 | 0.001; 0.005 |
| **Random effects** | | | | | | | | |
| Variance intercept | 2.639 | 0.296 | <0.0001 | | 3.854 | 0.406 | <0.0001 | |
| Variance slope | 0.018 | 0.006 | 0.001 | | 0.016 | 0.007 | 0.009 | |
| Cov. intercept, slope | −0.034 | 0.031 | 0.270 | | −0.066 | 0.039 | 0.093 | |
| Variance explained | 0.23 | | | 0.13 | | | | |

| | Concreteness | | | | Finitude | | | |
|---|---|---|---|---|---|---|---|---|
| | Estimate | SE | p | 95% CI | Estimate | SE | p | 95% CI |
| **Fixed effects** | | | | | | | | |
| Intercept | 7.777 | 0.102 | <0.0001 | 7.576; 7.978 | 7.270 | 0.123 | <0.0001 | 70.028; 7.511 |
| Age | −0.047 | 0.007 | <0.0001 | −0.062; −0.032 | 0.023 | 0.009 | 0.010 | 0.006; 0.041 |
| Time | −0.047 | 0.012 | <0.0001 | −0.071; −0.024 | 0.024 | 0.012 | 0.054 | −0.000; 0.048 |
| Age*Time | 0.001 | 0.001 | 0.486 | −0.001; 0.002 | 0.001 | 0.001 | 0.409 | −0.001; 0.003 |
| **Random effects** | | | | | | | | |
| Variance intercept | 2.953 | 0.339 | <0.0001 | | 4.863 | 0.477 | <0.0001 | |
| Variance slope | 0.008 | 0.006 | 0.090 | | 0.006 | 0.006 | 0.189 | |
| Cov. intercept, slope | −0.055 | 0.034 | 0.108 | | −0.092 | 0.041 | 0.026 | |
| Variance explained | 0.08 | | | | 0.05 | | | |

Chronological age was grand–mean–centered at 54.2 years.
CI confidence intervals.

**Table 3 | Growth Models of Life Satisfaction, with Time Perspectives as Predictors and (A) Age and (B) SRLE as Moderator**

| | (A) Estimate | SE | p | 95% CI | (B) Estimate | SE | p | 95% CI |
|---|---|---|---|---|---|---|---|---|
| **Fixed effects** | | | | | | | | |
| Intercept | 2.815 | 0.051 | <0.0001 | 2.715; 2.915 | 2.852 | 0.052 | <0.0001 | 2.750; 2.955 |
| Age | 0.010 | 0.001 | <0.0001 | 0.008;0.013 | | | | |
| SRLE | | | | | −0.007 | 0.001 | <0.0001 | −0.010; −0.005 |
| Gender | 0.058 | 0.034 | 0.0839 | −0.008; 0.124 | 0.058 | 0.035 | 0.100 | −0.011; 0.127 |
| Self-Rated Health | 0.168 | 0.020 | <0.0001 | 0.1305; 0.207 | 0.171 | 0.021 | <0.0001 | 0.1303; 0.212 |
| Income | 0.047 | 0.013 | 0.0003 | 0.022; 0.072 | 0.044 | 0.013 | 0.001 | 0.018; 0.070 |
| Education: Vocational College | 0.031 | 0.054 | 0.561 | −0.075; 0.138 | −0.022 | 0.055 | 0.685 | −0.130; 0.086 |
| Education: College | 0.011 | 0.065 | 0.871 | −0.117; 0.138 | −0.043 | 0.067 | 0.515 | −0.174; 0.087 |
| Education: University | −0.038 | 0.052 | 0.461 | −0.141; 0.064 | −0.085 | 0.053 | 0.113 | −0.190; 0.020 |
| bpConcreteness | 0.072 | 0.009 | <0.0001 | 0.0531; 0.090 | 0.066 | 0.010 | <0.0001 | 0.047; 0.085 |
| bpPast Orientation | −0.004 | 0.009 | 0.644 | −0.022; 0.014 | −0.002 | 0.010 | 0.863 | −0.021; 0.017 |
| bpObsolescence | −0.039 | 0.010 | 0.0003 | −0.060; −0.018 | −0.044 | 0.011 | <0.0001 | −0.066; −0.022 |
| bpFinitude | 0.004 | 0.007 | 0.619 | −0.011; 0.019 | 0.008 | 0.008 | 0.326 | −0.008; 0.023 |
| bpConcreteness*Age | −0.000 | 0.001 | 0.788 | −0.002; 0.001 | | | | |
| bpPast Orientation*Age | 0.000 | 0.001 | 0.762 | −0.001; 0.002 | | | | |
| bpObsolescence*Age | 0.000 | 0.001 | 0.513 | −0.001; 0.002 | | | | |
| bpFinitude*Age | −0.000 | 0.001 | 0.474 | −0.001; 0.001 | | | | |
| bpConcreteness*SRLE | | | | | 0.001 | 0.001 | 0.219 | −0.001; 0.002 |
| bpPast Orientation*SRLE | | | | | 0.000 | 0.001 | 0.720 | −0.001; 0.002; |
| bpObsolescence*SRLE | | | | | −0.000 | 0.001 | 0.9356 | −0.001; 0.001 |
| bpFinitude*SRLE | | | | | 0.000 | 0.001 | 0.381 | −0.001; 0.002 |
| Time | 0.001 | 0.003 | 0.8409 | −0.006; 0.007 | 0.001 | 0.003 | 0.788 | −0.005; 0.007 |
| wpConcreteness | 0.033 | 0.008 | <0.0001 | 0.017; 0.048 | 0.033 | 0.008 | <0.0001 | 0.018; 0.049 |
| wpPast Orientation | −0.011 | 0.007 | 0.126 | −0.026; 0.003 | −0.012 | 0.008 | 0.116 | −0.027; 0.003 |
| wpObsolescence | −0.016 | 0.008 | 0.057 | −0.032; 0.000 | −0.017 | 0.008 | 0.038 | −0.034; −0.000 |
| wpFinitude | 0.002 | 0.008 | 0.818 | −0.013; 0.017 | 0.001 | 0.008 | 0.859 | −0.014; 0.016 |
| wpConcreteness*Age | −0.001 | 0.001 | 0.167 | −0.002; 0.000 | | | | |
| wpPast Orientation*Age | 0.000 | 0.001 | 0.530 | −0.001; 0.001 | | | | |
| wpObsolescence*Age | −0.000 | 0.001 | 0.791 | −0.001; 0.001 | | | | |
| wpFinitude*Age | −0.000 | 0.001 | 0.847 | −0.001; 0.001 | | | | |
| wpConcreteness*SRLE | | | | | 0.000 | 0.001 | 0.736 | −0.001; 0.001 |
| wpPast Orientation*SRLE | | | | | 0.003 | 0.001 | 0.501 | −0.001; 0.001 |
| wpObsolescence*SRLE | | | | | −0.000 | 0.001 | 0.751 | −0.001; 0.001 |
| wpFinitude*SRLE | | | | | 0.000 | 0.001 | 0.917 | −0.001; 0.001 |
| **Random effects** | | | | | | | | |
| Variance intercept | 0.054 | 0.012 | <0.0001 | | 0.064 | 0.012 | <0.0001 | |
| Variance slope | 0.001 | 0.000 | 0.0002 | | 0.001 | 0.000 | 0.0002 | |
| Cov. intercept, slope | −0.000 | 0.002 | 0.931 | | −0.000 | 0.002 | 0.791 | |
| **Variance explained** | | | | | | | | |
| | 0.21 | | | | 0.21 | | | |

*N* = 420 who provided 1239 observations, bp = between–person, wp = within-person, CI = confidence intervals, Gender was coded as 0 = male, 1 = female, Self-rated health: Higher scores indicate better self-rated health, Education: reference group is vocational training. Income was rescaled in thousands so that a difference in income by one unit corresponds to a difference of 1000 Euro. Cov. = covariance. Unstandardized estimates and standard errors are presented. Chronological age was grand–mean–centered at 54.2 years, and subjective remaining life expectancy was grand-mean centered at 26.6 years.

effects indicated that overall higher concreteness and lower obsolescence were related to higher life satisfaction. From a within-person perspective, individuals reported higher life satisfaction on measurement occasions when they scored higher on concreteness. We did not find evidence that

these associations were moderated by age or SRLE. Looking at four satisfaction domains revealed that finitude and satisfaction with physical health were positively interrelated only for individuals with a higher SRLE. Associations of lower past-orientation and higher concreteness with a higher

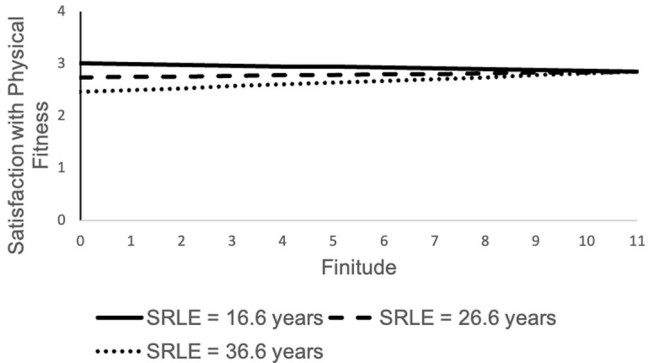

**Fig. 1 | Association of finitude (between-person component) with satisfaction with physical fitness by subjectively remaining life expectancy (SRLE).** *n* = 420.

satisfaction with mental fitness were moderated by age, with an effect of both time perspectives only on younger but not older adults' satisfaction.

### Time perspectives, life satisfaction, and age

Our results seem not entirely consistent regarding the relation between time perspective and well-being, nor the idea that age-related changes in time perspective are beneficial for well-being. We only found significant relations for two out of four assessed facets, moderating effects of age or SRLE were confined to two of four satisfaction domains, and overall, effects were small. Previous findings[15] that largely supported theoretical assumptions, however, seem questionable as they did not provide insights into the unique contributions of different time perspective facets for well-being. Unlike the previous study, we assessed time perspectives simultaneously within one model. Our analyses were based on a long-term observational period, an age-heterogeneous sample ranging from young to old age, and our analytical approach included an explicit differentiation of between- vs. within-person associations of time perspectives with life satisfaction.

While our findings support the idea that time perspective is a multi-dimensional construct[18,19] that shows multi-directional age-related changes, they do not speak for the idea that these changes affect specifically older adults' well-being. Gaining insights into how older adults manage and maintain their well-being could be facilitated by considering the broader literature on well-being, and the determinants postulated therein. Telic theories center on goals and goal pursuit as determinants of well-being[33]. To contribute to well-being, goals should match the person's motives and the individual's life context. The time perspective facet most consistently linked to life satisfaction in our study, concreteness, is also conceptually related to goals and goal pursuit. It, thus, seems worthwhile to investigate the unique contributions of concreteness for predicting well-being above and beyond more general motivational variables (e.g., self-efficacy) or personality traits (e.g., optimism). Similarly, one should determine whether age-related changes in action regulation, motives, or living conditions moderate or mediate the relation between concreteness and well-being.

### Implications

While theoretical approaches to time perspective provide insights into its link to well-being, theories focusing on well-being rarely elaborate on the role of time perspective (for an exception, see ref. 34). One reason why time perspective has received little attention in theories of well-being is that a plurality of well-being theories exists and these often remain separate and distinct from each other rather than being integrated into a more comprehensive conceptualization[35,36]. Given that human beings impose subjective, time-related interpretations on their existence[37,38], time perspective is an important construct for research on well-being and emotional aging[39]. The relevance of time perspective for well-being research could be increased by taking a closer look at the current operationalizations and measurements of time perspective. Specifically, there seems to be overlap on the conceptual

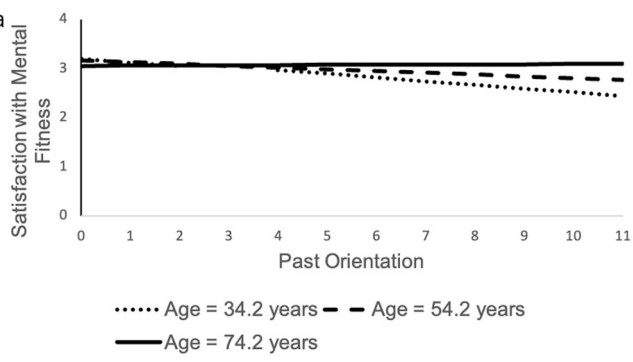

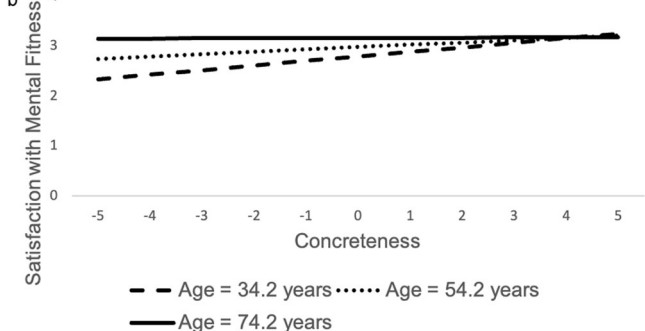

**Fig. 2 | Associations of time perspective and domain-specific satisfaction.** Association of (**a**) past orientation (between-person component) and (**b**) Concreteness (within-person component) with satisfaction with mental fitness by chronological age. *n* = 420.

level. For example, Brandtstädter's facet of affective valence, defined as optimistic outlook on the future, may not be distinguishable from dispositional optimism (generalized positive expectation for the future). Time perspective measures have also been criticized because items are not clearly and exclusively linked to the time perspective they are meant to assess[40,41]. Based on our findings, it is important to move towards multi-dimensional conceptualizations of time perspective, especially concerning SST which only considers future time, but also concerning SWB which, according to our findings, is only related to specific time perspective aspects such as concreteness or obsolescence. Incorporating a multi-dimensional approach will better capture the complex nature of developmental changes in time perspective and well-being. As age moderation effects of the relation between time perspective and life satisfaction were circumscribed in our study, theories of time perspective should reconsider the role of age in this relation. This could be done by specifying boundary conditions under which age-related changes in time perspective are not or no longer related to higher SWB (e.g., when approaching the end of life[37]). Overall, a more comprehensive understanding of the relation between time perspective and well-being might require more clarification, explication, comparison, and integration of the current conceptualization of time perspective.

### Limitations

We complemented previous work on the relation between time perspective and well-being by assessing qualitative aspects of time perspective, used 10-year longitudinal data of an age-diverse sample, and tested age and SRLE as moderators. Analyzing the associations of life satisfaction with different facets of time perspective simultaneously in one analysis, while controlling for relevant covariates allowed us to estimate the unique between- and within-person effects of time perspective. Despite these strengths, the following limitations deserve note. First, the AAF data set only included three measurement occasions. More occasions are necessary to investigate potential nonlinear changes. Three measurement occasions are the minimum for analyzing within-person

**Article**

predictor effects. With more measurement occasions, we might have found stronger effects of time perspectives, particularly on the within-person level. Given that the AAF data set included only a short version of the time perspective scale[9] that had moderate reliabilities, stronger and more consistent effects of time perspective could be obtained with a longer version and higher reliabilities. As indicator of SWB, we only included life and domain satisfaction which capture cognitive well-being[33]. Given the multidimensionality of SWB, time perspective might have different associations with affective well-being.

Second, our results do not allow us to draw inferences concerning the causal direction of effects. Although motivated by theoretical reasoning about the temporal order of the study variables, we cannot exclude that changes in life satisfaction resulted in changes in time perspective[42], or that there were reciprocal influences. Additional background variables that were not assessed or considered in our analyses (e.g., personality, objective health) may have influenced both time perspective and life satisfaction.

Third, the AAF data set includes few adults older than 80 years and most individuals still reported a SRLE of 10 years or more. It might be important to learn about time perspective and its impact on well-being in very old age when time is running out, both objectively and subjectively and the human system reaches its highest level of vulnerability[37].

## Conclusion
Using data from a 10-year longitudinal study and focusing on the unique contribution of each time perspective for explaining changes in SWB, we found only selective and small associations of time perspective facets with changes in life satisfaction. For general life satisfaction, there was no evidence that this relation was moderated by age or SRLE. Moderation effects were only found for two of four indicators of domain-specific satisfaction. However, these moderation effects were small in size. Research concerning the relation between time perspective and well-being would profit from clarification and explication of concepts, investigations of mechanisms underlying the association as well as attempts at integrating both lines of research.

## Data availability
The datasets analyzed during the current study are not publicly available because participants of the AAF study have not consented to their data being made publicly available. Data is available from the corresponding author (Maria Wirth) on request. Data access can be obtained for scientific purposes only, which are carried out at academic institutions.

## Code availability
The SAS code for the main analyses is available at https://osf.io/4gwby. SAS 15.3 and PROC MIXED[28] were used to compute the longitudinal multilevel regression models.

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

## Acknowledgements

This work was supported by several grants of the VolkswagenStiftung (Az. II/83 142, Az. 86 758, Az. 93 272) awarded to K. Rothermund. The funder had no role in the conceptualization, design, data collection, analysis, decision to publish, or preparation of the manuscript.

## Author contributions

Maria Wirth assisted in interpreting the data, wrote the first draft, and critically edited the manuscript. Markus Wettstein analyzed and interpreted the data, and critically edited the manuscript. Klaus Rothermund conceptualized and designed the study, assisted in the interpretation of the data as well as critically edited the manuscript.

## Funding

## Competing interests

The authors declare no competing interests.
