## [Peer Review File · Communications Psychology]

2nd Apr 24

Dear Dr Wirth,

Thank you for your patience during the peer-review process. Your manuscript titled "Longitudinal Associations Between Time Perspective and Life Satisfaction Across Adulthood" has now been seen by 3 reviewers, and I include their comments at the end of this message. They find your work of interest but raised some important points. We are interested in the possibility of publishing your study in *Communications Psychology*, but would like to consider your responses to these concerns and assess a revised manuscript before we make a final decision on publication.

We therefore invite you to revise and resubmit your manuscript, along with a point-by-point response to the reviewers. Please highlight all changes in the manuscript text file.

Editorially, we consider it important that the revised manuscript provides a better introduction and discussion of the dimensions of time perspective, their operationalization (including the limitations of the measurement approach), and their theoretical relationship with well-being/life-satisfaction.

Please ensure you follow our statistical guidelines when reporting statistics (<https://www.nature.com/commpsychol/submit/submission-guidelines#statistical-guidelines>). Please note in particular our requirements for the reporting and interpretation of null-results. Non-significant findings derived from null-hypotheses significance tests should be reported in full, but may not be interpreted. Where you interpret null results, this interpretation must be based on Bayes Factors or equivalence tests.

I am attaching an Editorial Requests Table that details critical reporting requirements for the revised manuscript. Please attend to each item and ensure your manuscript is fully compliant. We are requesting that your manuscript aligns with these requirements as this facilitates the evaluation of your manuscript, reducing delays in re-review and potential future acceptance. If your revised manuscript is not aligned with these requests on major issues, such as those concerning statistics, it may be returned to you for further revisions without re-review. Additional information can be found in our style and formatting guide *Communications Psychology* formatting guide.

Please use the following link to submit your

- revised manuscript,
- point-by-point response to the referees' comments,
- cover letter (as a separate document),
- the Editorial Policy Checklist (see below),
- the Reporting Summary (see below), and
- the completed Editorial Request Table (attached):

[link redacted]

Best regards,

Jennifer Bellingtier

Jennifer Bellingtier, PhD

Senior Editor

Communications Psychology

REVIEWER EXPERTISE:

Reviewer #1 lifespan development, subjective perceptions of aging

Reviewer #2 lifespan development, subjective perceptions of aging

Reviewer #3 lifespan development, subjective perceptions of aging

REVIEWER REPORTS:

Reviewer #1 (Remarks to the Author):

The authors did a great job of unpacking an issue about time perspective and well-being and using state-of-the-art methods to analyze it. I don't see a flaw in the argument or the analyses. I am, though, not really sure what the manuscript actually tells me. The issues I am wondering about are more on a meta-level that seems outside of what can be addressed/revised (see below).

- A concrete suggestion is to define the dimensions of time perspective early in the manuscript. They are just named, but not described what they are about, such as what is obsolescence. I am wondering about two things here: 1) I am somewhat surprised about the example item for obsolescence. The example item seems to be more about "generational contempt" (for the lack of a better word) than about feelings of being out-of-touch, being outdated, etc. 2) "Concreteness" sounds confusing to me; "future orientation" might be better wording or using the whole phrase "concreteness of future perspective".

- On a broader meta-level, I have a hard time to see all of these dimensions to be really about time perspective. Obviously, they were published under that umbrella, but they don't make a lot of sense to me.

If one looks at the scales, the concreteness of future perspective is about making goals. (There is also a good bit of optimism in the items.) Goals are obviously in the future, but does that mean that all goal-related scales are actually time perspective measures? This is like saying conscientiousness is a time perspective construct because people answer items about

punctuality. Neuroticism is a time perspective measure because people are anxious about what will happen (in the future).

Obsolescence is a strange dimension; that seems to be a mix of pessimism, nostalgia, and old-men-yelling-at-clouds constructs. Even looking at other items of the scale: "Ich bin voll auf der Höhe der Zeit". I don't know whether the authors used that item, but it is not about time per se but about being at peak performance, which can be interpreted in several ways.

Finitude of life seems to be a death attitude; and there are a good bunch of different death attitudes that - I guess - would all be considered to be time perspectives... but it is not about time, it is about acceptance of death.

The only scale that I see has some real time perspective in it would be the past orientation. But past orientation is the one that has the cleanest relationship with age as you would expect that someone with more years can also look back at more experiences.

So, to say that differently: I am not convinced that future perspective, obsolescence, and finitude of life as measured by the scale are really time perspective dimensions. And to be clear, that criticism is not limited to the TPQ. I have the same concern about the Zimbardo time perspective scale; it is probably worse where I would think it measures every kind of personality construct but hardly anything about time. I think the authors acknowledge that fact themselves that the construct space of a lot of these time perspective measures seems to be muddy.

But I am also saying that because the two dimensions that were actually related to life satisfaction, future perspective and obsolescence, are - in my opinion - not about time, which makes it even worse for the idea that time perspective is related to life satisfaction.

I am not sure whether the authors would agree with my opinion, but I would invite the authors to consider adding something about the actual meaning of these dimensions and whether they actually assess time perspective.

- The other issue I am wondering about - which is probably more like a comment than a question - is the relationship between the internal consistencies and the retest correlations. The internal consistencies are rather low - even for 3 items. However, the retest correlations are nearly as high, which is really surprising. That seems to be an astonishing high stability if nearly all the reliable variance is also stable.

- Table 2 reports the means for the scales. Based on the method section, the time perspective items were assessed on a 1-to-5 scale and summed up. That would suggest means between 3 and 15 (for three items). But obsolescence at T1 is reported to be 2.88. Is that value correct or was the

scale actually 0 to 4 as the other scales? Though, in general, the means for obsolescence seems to be very low. I assume that there are a bunch of people who reported 1 (or 0) throughout.

Daniel Grün

Reviewer #2 (Remarks to the Author):

The present manuscript describes a study that examined age-related changes in time perspective (past-orientation, concreteness of the future, obsolescence, and finitude) using data from an age-diverse sample of 459 adults in the “Aging as Future” study spanning 10 years. Results showed that those with higher concreteness and lower obsolescence reported higher life satisfaction, and those who with higher concreteness of the future also reported higher life satisfaction on measurement occasions. The associations were not moderated by age, raising theoretical questions about age-differential associations between time perspective and well-being.

The study question is interesting in light of gaps in the current literature on the subjective experience of time and related correlates. The study design is apt, as are the measures, sample, and general approach to the data analyses.

The authors may wish to consider the following points to hone their manuscript.

The introduction offers a literature review focused on relationships between time perspectives and well-being. The present study focuses on life satisfaction, per se. Can the authors say more about why they selected life satisfaction as their indicator of well-being, and if it has been used in the previous research that was discussed? This might have some bearing on the conclusion that results “seem not entirely consistent regarding the relation between time perspective and well-being.”

Although a multidimensional measure of life satisfaction across 9 domains was used, only overall scores were analyzed. This seems like a missed opportunity – even though the averaged single indicator that was computed has good internal consistency. Can more be said about the decision to use only a general measure of life satisfaction as an indicator of well-being? And/or perhaps in the discussion comment on if looking at individual domains might be of theoretical interest.

It would be useful to know the means for the main study variables (life satisfaction, time perspectives) across the 5 age groups, by time. This could be a table or a figure. Table 2 only presents the overall means.

I'm having a bit of trouble deciphering the observed age effects. For example, several effects are noted for "chronologically older adults". Does this mean that the older age group differed from the others, or that there was an association with advancing age? A bit more description about the age associations that were found would be useful.

As well, it is noted that associations between time perspectives and life satisfaction were similar in size among young, middle-aged, and older adults. How do these effects align with the 5 age groups that were sampled?

Additionally, it was also reported that older adults reported higher levels of obsolescence, past-orientation, and finitude but lower levels of concreteness at T1. Was this just for individuals in the 1929-1938 cohort?

The item example reflecting feelings of obsolescence (e.g., "I have increasingly less sympathy for the views of the younger generation") seems to be tapping more of a negative age attitude as opposed to the sense that one is becoming outdated. A different item might work better as an example.

In summary, the study addresses a standing question in a unique way with the use of multiple time perspective measures and age groups. Contrary to prior research and theoretical predications, limited relationships were observed between time perspectives and life satisfaction. While the study offers a good view of age-related changes/stability in time perspectives, more could be done to say how this work advances a theoretical understanding with respect to well-being.

Reviewer #3 (Remarks to the Author):

The manuscript describes a study investigating the 10-year longitudinal relationship between time perspective (assessed with four sub-scales) and life satisfaction.

The manuscript is well-written and clear. The authors also considered the moderating roles of age and subjective remaining life expectancy. Multilevel regression models were used, and data was analysed at the within and between person level. The authors found that participants with overall higher scores on the time perspective scale “concreteness” and lower scores on the time perspective scale “obsolescence” reported greater life satisfaction. Participants also reported greater life satisfaction on measurement occasions when they scored higher on the time perspective scale “concreteness”. Contrary to what hypothesized, age and subjective remaining life expectancy did not moderate the investigated relationships. Overall, the manuscript is well written, clear to follow, and the analyses are sound. Strengths of the study are the use of a multi-dimensional measure of time perspective (comprising 4 scales); the 10-year follow-up; the wide age range of participants which allowed the authors to test the moderating role of age; and the conduct of analyses at the within- and between- person level. Perhaps a limitation of the study is that self-rated health, rather than objectively assessed health, is used as a covariate.

I have only minor comments on the manuscript:

- Abstract: I assume this is not possible due to limited wordcount but to me from the abstract it is not clear what the “concreteness” and “obsolescence” scales measure and hence to fully understand the results. A brief definition in the background section or an example item would help. This applies also to the introduction of the manuscript.
- Line 70: could the authors provide some examples/references of studies that did not include younger and middle-aged adults and of studies with short time periods. Alternatively, could the authors cite a review, if available.
- Line 73: same comment. Could the authors provide some references.
- Page 5, line 92: could the authors please define the sub-categories of time perspective.
- Covariates: why did the authors not controlled for mental health or depression? Is this because of similarity with wellbeing as outcome?

Also, is lack of objective health as covariate a limitation or did the authors purposefully chosen self-rated health because of its subjectivity?

We ask that you ensure your manuscript complies with our editorial policies and reporting requirements.

To that end, we require revised manuscripts to be accompanied by two completed items: a reporting summary that collects information on study design and procedure, and an editorial policy checklist that verifies compliance with all required editorial policies.

Nature Research Reporting Summary

Editorial Policy Checklist

All points on the policy checklist must be addressed. Your revised manuscript can only be sent back to the referees if these checklists are completed and uploaded with the revision.

Notes: If you have submitted a Stage 1 Registered Report, Review, Primer, Comment, or Perspective you do not need to submit these forms. If you have already submitted these forms, you may disregard this request.

Comments Reviewer 1:

1. *The authors did a great job of unpacking an issue about time perspective and well-being and using state-of-the-art methods to analyze it. I don't see a flaw in the argument or the analyses. I am, though, not really sure what the manuscript actually tells me. The issues I am wondering about are more on a meta-level that seems outside of what can be addressed/revise (see below).*

Thank you for your positive appraisal of the manuscript. As you will see in the revised manuscript and detailed in our answers below, we have tried to attend to your concerns regarding the conceptualization and assessment of time perspective and point out the novelty of our contribution. We have also corrected the mistake regarding the scale range of the time perspective scales.

2. *A concrete suggestion is to define the dimensions of time perspective early in the manuscript. They are just named, but not described what they are about, such as what is obsolescence. I am wondering about two things here: 1) I am somewhat surprised about the example item for obsolescence. The example item seems to be more about “generational contempt” (for the lack of a better word) than about feelings of being out-of-touch, being outdated, etc. 2) “Concreteness” sounds confusing to me; “future orientation” might be better wording or using the whole phrase “concreteness of future perspective”.*

We agree that it is helpful to define the time perspective dimensions as conceptualized by Brandtstädter early on. We have now added a description of these dimensions as put forward by Brandtstädter and colleagues (1994, 1997) on p. 4. Additionally, we have changed the obsolescence item to one that better represents the concept of obsolescence (“I increasingly have the feeling that I have lost touch with modern times”). Given the names of the time perspective facets such as “concreteness of future time”, we are bound to the names used in previous publications by Brandtstädter. Additionally, to avoid confusion with time perspective concepts established by other researchers, we maintain using the original name “concreteness of future time”. In most parts of the manuscript, we use the facet’s full name and only in a few parts of the manuscript, we shortened it to “concreteness”. This was done due to word restrictions.

3. *On a broader meta-level, I have a hard time to see all of these dimensions to be really about time perspective. Obviously, they were published under that umbrella, but they don't make a lot of sense to me. If one looks at the scales, the concreteness of future perspective is about making goals. (There is also a good bit of optimism in the items.) Goals are obviously in the future, but does that mean that all goal-related scales are actually time perspective measures? This is like saying conscientiousness is a time perspective construct because people answer items about punctuality. Neuroticism is a time perspective measure because people are anxious about what will happen (in the future). Obsolescence is a strange dimension; that seems to be a mix of pessimism, nostalgia, and old-men-yelling-at-clouds constructs. Even looking at other items of the scale: "Ich bin voll auf der Höhe der Zeit". I don't know whether the authors used that item, but it is not about time per se but about being at peak performance, which can be interpreted in several ways. Finitude of life seems to be a death attitude; and there are a good bunch of different death attitudes that - I guess - would all be considered to be time perspectives... but it is not about time, it is about acceptance of death. The only scale that I see has some real time perspective in it would be the past orientation. But past orientation is the one that has the cleanest relationship with age as you would expect that someone with more years can also look back at more experiences. So, to say that differently: I am not convinced that future perspective, obsolescence, and finitude of life as measured by the scale are really time perspective dimensions. And to be clear, that criticism is not limited to the TPQ. I have the same concern about the Zimbardo time perspective scale; it is probably worse where I would think it measures every kind of personality construct but hardly anything about time. I think the authors acknowledge that fact themselves that the construct space of a lot of these time perspective measures seems to be muddy.*

We see that there can be different opinions on whether the concepts and facets subsumed under the umbrella term of time perspective are about time per se and how they relate to other psychological constructs. Generally, time perspective is conceptualized as a malleable, cognitive-motivational construct that develops and changes as a function of experience over the lifespan. Importantly, time perspective as proposed by Brandtstädter captures various facets that describe how time is experienced by an individual, and this broad conception comprises attitudes and relations towards the past and the future, towards the finitude of an individual's lifetime, and also an individual's experience of and feelings towards "changing times". Time perspective, thus, captures whether the past and future

provide sources of meaning, whether finitude undermines meaningfulness or can be transcended, and whether an individual feels integrated or estranged from societal and historical changes. Thus, rather than focusing on a narrow conception of “time per se”, time perspective as proposed by Brandtstädter, considers various aspects of individual and historical time, and how these temporal changes are experienced by individuals. Although these facets may have relations to and implications for other constructs (e.g., personality, motivation, or affective experience), the time perspective scales are defined in terms of an individual’s experience of different facets of individual and historical time, which is covered by the wording and meaning of the scale items. Time perspective is also conceptually and empirically distinct from personality traits, motivation, and affect, although they are meaningfully related (Koji et al., 2018; Webster et al., 2014). This, of course, does not exclude that there is a certain overlap in the content of constructs or in the way time perspective and personality traits are assessed. The different measures of time perspective have been criticized as not being clearly and exclusively linked to the time perspective they are meant to assess (Chu & Carstensen, 2023; Webster, 2011). We now elaborate on this issue in our discussion (pp. 17-18).

At the same time, we would like to point out that our analyses are based on concepts established in research on time perspective. The time perspective concepts of Brandtstädter and colleagues comprise qualitative aspects and are rather broad, thus, potentially relating to other constructs such as optimism (specifically the affective valence of the future scale defined as optimistic expectations about the future). We now acknowledge that research on time perspective has to clearly define and explain what is meant by the constructs and which facet of time and which type of relation is covered by the different constructs. It will also be important to establish incremental validity for the assessed facets such as concreteness over and above, for example, variables relating to goal pursuit.

Concerning past orientation, we agree that older adults have more years to look back on, however, as shown in previous studies, this alone does not mean that older compared to younger adults’ daily lives and thoughts are pre-occupied with the past (Grühn et al., 2016; Hepper et al., 2020).

4. *But I am also saying that because the two dimensions that were actually related to life satisfaction, future perspective and obsolescence, are - in my opinion - not about time, which makes it even worse for the idea that time perspective is related to life satisfaction.*

We would like to point out that our analyses are based on concepts established in research on time perspective (Brandtstädter & Wentura, 1994; Brandtstädter et al., 1997; Rothermund & Brandtstädter, 2003). The time perspective concepts of Brandtstädter and colleagues comprise qualitative aspects and are rather broad (see our response to your comment 3). We now acknowledge that research on time perspective and well-being has to clearly explain how the variables relate to different facets of time and how these facets are experienced and also test the incremental validity of the used measures. (Please also see our response to your comment 3).

5. *I am not sure whether the authors would agree with my opinion, but I would invite the authors to consider adding something about the actual meaning of these dimensions and whether they actually assess time perspective.*

We have added some ideas and clarifications on how the scales capture time perspective, how time perspective could be situated within the well-being literature, and what progress both in terms of conceptualization and assessment is necessary to move the field forward (Please also see our response to your comment 3).

6. *The other issue I am wondering about - which is probably more like a comment than a question - is the relationship between the internal consistencies and the retest correlations. The internal consistencies are rather low - even for 3 items. However, the retest correlations are nearly as high, which is really surprising. That seems to be an astonishing high stability if nearly all the reliable variance is also stable.*

The internal consistencies obtained in our study (.60 - .77) are comparable to those that were reported in previous studies (.70-.74) using all items for the investigated scales (Brandtstädter & Wentura, 1994; Brandtstädter et al., 1997). One reason for this modest internal consistency could be that the scale items assess different (broader) aspects of each time perspective dimension, thus, the item content likely is not very homogenous. How individuals answer these items across time, however, seems to be rather consistent as signified in the high stability found in our study (.51-.64) but also in the previous

longitudinal study by Brandtstädter and colleagues (.58-.69). Cronbachs alpha has been criticized for different reasons (Sijtsma, 200), and one of its limitations is that it might underestimate the reliability of a scale when the item content of the scale is heterogeneous. Therefore, the alpha scores we obtained might rather represent a conservative estimate of reliability, and the re-test correlations as an alternative reliability estimate indicate that the time perspective scales reveal adequate reliability, although scales with more items would have additionally reduced measurement error.

7. *Table 2 reports the means for the scales. Based on the method section, the time perspective items were assessed on a 1-to-5 scale and summed up. That would suggest means between 3 and 15 (for three items). But obsolescence at T1 is reported to be 2.88. Is that value correct or was the scale actually 0 to 4 as the other scales? Though, in general, the means for obsolescence seems to be very low. I assume that there are a bunch of people who reported 1 (or 0) throughout.*

Thank you for pointing out this mistake. The scale range was 0-4. We have now corrected this in our manuscript. Concerning the individuals who scored the lowest obsolescence values – between 9-11% of participants per measurement occasion reported the lowest value. However, across all measurement points, only 15 participants consistently scored 0. These results indicate that there were no floor effects for the scales and that although some individuals started with the lowest value, change in obsolescence was observed in most of these individuals. The intraclass correlation coefficient for obsolescence was $r = .50$, which means that 50% of the overall variation in obsolescence was due to interindividual differences, and 50% was due to intraindividual variability. Our analysis of obsolescence trajectories (Table 2) resulted in significant random effects of both intercept and slope so that there is substantial interindividual variation both in obsolescence baseline scores as well as in obsolescence changes over time.

Comments Reviewer 2:

1. *The introduction offers a literature review focused on relationships between time perspectives and well-being. The present study focuses on life satisfaction, per se. Can the authors say more about why they selected life satisfaction as their indicator of well-being, and if it has been used in the previous research that was discussed? This might have some bearing on the conclusion that results “seem not entirely consistent regarding the relation between time perspective and well-being.”*

Thank you for pointing out that we need to be more specific in terms of the predicted outcome. We chose life satisfaction as an indicator of subjective well-being for three reasons:

- 1) For practical reasons as life satisfaction was the only indicator of well-being that has been continuously assessed in the AAF project.
- 2) Based on the criticism that common measures of the affective component of subjective well-being (e.g. PANAS) do not assess affect regarding one’s life but rather affect unrelated to one’s life as a whole (e.g., Simsek, 2008).
- 3) None of the previous longitudinal studies investigating the relation between time perspective and well-being has looked into life satisfaction as an outcome. Most previous studies used an indicator of the affective dimension of subjective well-being and one study focused on psychological well-being.

We now state our motivation to focus on life satisfaction as a subjective well-being indicator more clearly in the introduction on page 7.

2. *Although a multidimensional measure of life satisfaction across 9 domains was used, only overall scores were analyzed. This seems like a missed opportunity – even though the averaged single indicator that was computed has good internal consistency. Can more be said about the decision to use only a general measure of life satisfaction as an indicator of well-being? And/or perhaps in the discussion comment on if looking at individual domains might be of theoretical interest.*

We initially chose to focus on a general/ aggregated measure of life satisfaction to keep the method and result sections tangible. Presenting the analyses for every satisfaction domain would have been too complex. However, considering that time perspective is conceptualized as multidimensional and given previous multidirectional age-related

changes in this construct, it seems warranted to also employ a multidimensional approach to the outcome. To keep the method and result section at a reasonable length, we now explore the effects of time perspective for satisfaction dimensions that are most important across the lifespan and which were also available in the AAF data set. Thus, we report on analyses with satisfaction in the health, friends, physical and mental fitness domain. For satisfaction with mental fitness, we find two age-differential associations, indicating that time perspective (orientation towards the past and concreteness of future time perspective) is less strongly related to chronologically older adults' satisfaction compared to chronologically younger persons. We also found a moderating effect of SRLE, indicating that the positive effect of finitude on satisfaction with physical fitness is stronger in individuals with a higher SRLE. For more details, please see pp. 15-16.

3. *It would be useful to know the means for the main study variables (life satisfaction, time perspectives) across the 5 age groups, by time. This could be a table or a figure. Table 2 only presents the overall means.*

We have now added a table to the Appendix of our manuscript that includes the main study variables for each cohort and all measurement points.

4. *I'm having a bit of trouble deciphering the observed age effects. For example, several effects are noted for "chronologically older adults". Does this mean that the older age group differed from the others, or that there was an association with advancing age? A bit more description about the age associations that were found would be useful.*

We now see that it did not become very clear that the reported results are based on analyses using chronological age as a continuous variable rather than age cohort. This approach is more appropriate as specifying age groups is less accurate and does not make sufficient use of the available age information; there could, for instance, be differences within one specified age group that cannot be detected based on such a grouping approach. We now stress this in our method and results section. To further avoid the impression that we investigated age groups rather than chronological age, we have moved the table containing demographic information by age cohort to the Appendix.

5. *As well, it is noted that associations between time perspectives and life satisfaction were similar in size among young, middle-aged, and older adults. How do these effects align with the 5 age groups that were sampled?*

We see that our statement may imply that we used age cohort as a predictor in our analyses. However, we actually used chronological age as a continuous predictor (see our comment above) and our statement that the associations were similar across young, middle-aged, and older adults was based on our results showing no significant, moderating effect of chronological age. We have now deleted this statement.

6. *Additionally, it was also reported that older adults reported higher levels of obsolescence, past-orientation, and finitude but lower levels of concreteness at T1. Was this just for individuals in the 1929–1938 cohort?*

We now see that it did not become very clear that the reported results are based on analyses using chronological age rather than age cohort as a predictor. We now stress this in our method and results section. We also point out that the effects of chronological age are visualized in Figure 1.

7. *The item example reflecting feelings of obsolescence (e.g., “I have increasingly less sympathy for the views of the younger generation”) seems to be tapping more of a negative age attitude as opposed to the sense that one is becoming outdated. A different item might work better as an example.*

Thank you for pointing this out. We now present an example item that better portrays the dimension of obsolescence (“I increasingly have the feeling that I have lost touch with modern times”).

8. *In summary, the study addresses a standing question in a unique way with the use of multiple time perspective measures and age groups. Contrary to prior research and theoretical predications, limited relationships were observed between time perspectives and life satisfaction. While the study offers a good view of age-related changes/stability in time perspectives, more could be done to say how this work advances a theoretical understanding with respect to well-being.*

We agree that more could have been said about how our results add to the theoretical understanding and conceptualization of well-being. While theoretical approaches to time perspective provide insights into its link to well-being, theories focusing on well-being rarely elaborate on the role of time perspective (for an exception, see Durayappah, 2011). One reason why time perspective has received little attention in theories of well-being is that a plurality of well-being theories exists and these often remain separate and distinct from each other rather than being integrated into a more comprehensive conceptualization (Das et al., 2020; Feist et al., 1995; Linto et al., 2016). We now discuss the closeness in conceptualization of (telic) well-being theories and the time perspective facet concreteness as well as the fact that for time perspective concepts to become more relevant to well-being research, a closer look should be taken at whether the constructs and measurements really are related to time. For more details please also see pp. 17-18.

Comments Reviewer 3:

1. *Abstract: I assume this is not possible due to limited wordcount but to me from the abstract it is not clear what the “concreteness” and “obsolescence” scales measure and hence to fully understand the results. A brief definition in the background section or an example item would help. This applies also to the introduction of the manuscript.*

Yes, unfortunately, due to word constraints, we are unable to define the constructs “concreteness” and “obsolescence” in the abstract. We now define the time perspective concepts as proposed by Brandtstädter and colleagues in the introduction of the manuscript on p. 4.

2. *Line 70: could the authors provide some examples/references of studies that did not include younger and middle-aged adults and of studies with short time periods. Alternatively, could the authors cite a review, if available.*

We now specifically state which of the longitudinal studies investigating changes in the relation between time perspective and well-being did not include younger and/or middle-aged adults, and which only assessed the relation over a short time period. Unfortunately, we are not aware of a review that focuses specifically on longitudinal studies investigating the relation between time perspective and well-being.

3. *Line 73: same comment. Could the authors provide some references.*

We have now added some references for our statement that most studies on the relation between time perspective and well-being focus on chronological age.

4. *Page 5, line 92: could the authors please define the sub-categories of time perspective.*

We now define the time perspective concepts as proposed by Brandtstädter and colleagues on p. 4.

5. *Covariates: why did the authors not controlled for mental health or depression? Is this because of similarity with wellbeing as outcome?*

Depression is the only mental health indicator that was assessed in the AAF project (T1 only). Depression and our life satisfaction indicator are highly negatively correlated ($r = -$

.67). Thus, we did not include depression as a covariate due to its high overlap with life satisfaction.

6. *Perhaps a limitation of the study is that self-rated health, rather than objectively assessed health, is used as a covariate. Also, is lack of objective health as covariate a limitation or did the authors purposefully chosen self-rated health because of its subjectivity?*

Unfortunately, within the AAF project, participants' physical/objective health was not assessed, thus, we are not able to include this variable as a covariate in our analyses.

Previous studies point to either no relation between objective health and time perspective or low to medium correlations (Brandtstädter et al., 1997; Brandtstädter & Wentura, 1994). We have now added objective health to the variables that were not assessed and could have potentially influenced time perspective and life satisfaction as well as their interrelations (p. 19).

28th Jun 24

Dear Dr Wirth,

Your manuscript titled "Longitudinal Associations Between Time Perspective and Life Satisfaction Across Adulthood" has now been seen by our reviewers, whose comments appear below. In light of their advice I am delighted to say that we are happy, in principle, to publish a suitably revised version in *Communications Psychology* under the open access CC BY license (Creative Commons Attribution v4.0 International License).

We therefore invite you to revise your paper one last time to address the remaining concerns of our reviewers and a list of editorial requests. At the same time we ask that you edit your manuscript to comply with our format requirements and to maximise the accessibility and therefore the impact of your work.

EDITORIAL REQUESTS:

SUBMISSION INFORMATION:

OPEN ACCESS:

Communications Psychology is a fully open access journal. Articles are made freely accessible on publication under a CC BY license (Creative Commons Attribution 4.0 International License). This

license allows maximum dissemination and re-use of open access materials and is preferred by many research funding bodies.

For further information about article processing charges, open access funding, and advice and support from Nature Research, please visit <https://www.nature.com/commspsychol/article-processing-charges>

At acceptance, you will be provided with instructions for completing this CC BY license on behalf of all authors. This grants us the necessary permissions to publish your paper. Additionally, you will be asked to declare that all required third party permissions have been obtained, and to provide billing information in order to pay the article-processing charge (APC).

* **DATA AVAILABILITY:**

[link redacted]

Best regards,

Jennifer Bellingtier

Jennifer Bellingtier, PhD

Senior Editor

Communications Psychology

REVIEWERS' EXPERTISE:

Reviewer #1 lifespan development, subjective perceptions of aging

Reviewer #2 lifespan development, subjective perceptions of aging

REVIEWERS' COMMENTS:

Reviewer #1 (Remarks to the Author):

The authors did a great job in improving on the provided comments. I am still not convinced that some of the scales used are actually assessing time perception, but who cares...

I have several points primarily for improving the current manuscript with some constructive comments:

-- Paragraphs. Some paragraphs are very short resulting in a fuzzy look for the final document. And as my mentor said, a paragraph needs at least two sentences, I would suggest removing some paragraph breaks. (This is just a visual cosmetic suggestion!)

-- Include a more systematic literature on time perception and SWB. There are several studies that looked at time perception and measures of subjective well-being that haven't been discussed. I would suggest including a page for including cross-sectional and longitudinal studies. I assume the authors mainly ignored cross-sectional studies as they focus on the longitudinal pattern, but the findings from the cross-sectional studies (e.g. Gruhn et al., 2014) is mainly consistent with the longitudinal pattern. Moreover, I did a very rough search (not exhaustive at all) and some other longitudinal studies popped up that might be worth including (I did not vet these papers!):

- Lu (2022) - 10.1007/s11136-022-03163-6
- Fookien (1982) - 10.1177/016502548200500306
- Hill et al. (2022) - 10.1037/pag0000647 (big five and ftp?)
- Korff & Biemann (2020) - 10.1037/pag0000513

-- Rescaling variables. For the reporting of the model data, it might be advisable to rescale variables, such as income, age, and time. In particular income is suboptimal: Reporting a significant income effect as .000 with an SE of .000 and a CI of .000 to .000 is somewhat useless. Thus, I would suggest dividing income by 10'000 as this is usually done. Age and time might be less problematic, but I can see that it would help – especially for the age interactions to report it in decades rather than years.

This would have the additional benefit that the authors could then just report two decimal digits rather than three in their tables. That would tremendously clean-up the tables and make them easier to navigate.

-- Tables. I think there are too many tables. This is somewhat overkill. If there is a way to put tables 4 to 7 maybe in the appendix / online supplement, I think that would help. Or if there is a way to reduce these tables in a more compact or condensed form, that would help.

-- Figures. The figures seem to be all over the place. They have different styles, different fonts, different font sizes, different line widths, etc. Figure 2 has resolution issues, but I am also not sure that figure is actually needed. I don't think people really get that figure and I would rather remove it – or place it in the online supplement. I am even not sure about Figure 1 as the additional benefit over the table details is somewhat meek. Anyway, I would strongly suggest to redo the figures in a coherent and readable style (some of the font sizes are probably too small).

-- Conclusions. I would like to suggest to include a stronger discussion for the implications and conclusions of the findings. For example, what is the implication for SST or other concepts in the literature of the findings.

Reviewer #2 (Remarks to the Author):

This is my second review of the manuscript which describes a study that explored age-related changes in time perspective (past-orientation, concreteness of the future, obsolescence, and finitude) using longitudinal data from an age-diverse sample of 459 adults in the Aging as Future study.

The authors have done an apt and satisfactory job of addressing the key concerns of reviewers (several of which were common across reviewers). In particular, a better case has been made for why the present construct reflects “time perspective” (in light of previous research and theory), and how the work aligns with well-being given that only one measure of life satisfaction was used.

Additional information has been provided about the measures and sample as requested, and it has been made more clear how chronological age was used in the analysis (i.e., as a continuous variable, not reflecting distinct age groups) and confusions regarding the reporting of “age group” effects were remedied. Also, some attention has now been given the multidimensional nature of the life satisfaction variable.

The additional information added to the Implications sections helped to attenuate my concerns about what these results add to the broader literature.

In summary, the present study addresses a standing question in an informative way with the use of multiple time perspective measures and age groups. Contrary to prior research and theoretical predications, limited relationships were observed between time perspectives and life satisfaction. As such, the present findings offer new information about associations between time perspective and life satisfaction during adulthood.

Comments Reviewer 1:

1. *Paragraphs. Some paragraphs are very short resulting in a fuzzy look for the final document. And as my mentor said, a paragraph needs at least two sentences, I would suggest removing some paragraph breaks. (This is just a visual cosmetic suggestion!)*

Thank you for this suggestion. We have now considerably reduced the number of paragraphs.

2. *Include a more systematic literature on time perception and SWB. There are several studies that looked at time perception and measures of subjective well-being that haven't been discussed. I would suggest including a page for including cross-sectional and longitudinal studies. I assume the authors mainly ignored cross-sectional studies as they focus on the longitudinal pattern, but the findings from the cross-sectional studies (e.g. Gruhn et al., 2014) is mainly consistent with the longitudinal pattern. Moreover, I did a very rough search (not exhaustive at all) and some other longitudinal studies popped up that might be worth including (I did not vet these papers!):*

- Lu (2022) - 10.1007/s11136-022-03163-6
- Fookan (1982) - 10.1177/016502548200500306
- Hill et al. (2022) - 10.1037/pag0000647 (big five and ftp?)
- Korff & Biemann (2020) - 10.1037/pag0000513

We fully agree that, a more expansive review of the literature on time perception and SWB would be desirable, and we are very grateful for the suggested papers. However, given the space limitations of the manuscript, we see ourselves unable to do so. We now explicitly state that we focus our literature review exclusively on longitudinal studies that tested whether the relation between change in time perception and SWB was moderated by age, which is the main focus of the manuscript. We have now added a reference to two recent meta-analyses (Kooji et al., 2018 and Laureiro-Martinez et al., 2017) using cross-sectional data to elucidate the relation between time perspective and SWB and the role of age. The longitudinal studies mentioned above, unfortunately, did not test an age moderation of the relation between time perspective and SWB, did not include a measure of SWB, or used SWB as a predictor rather than the outcome. We, however, now cite Korff & Biemann (2020) when discussing the directionality of the relations between time perspective and SWB.

3. *Rescaling variables. For the reporting of the model data, it might be advisable to rescale variables, such as income, age, and time. In particular income is suboptimal: Reporting a significant income effect as .000 with an SE of .000 and a CI of .000 to .000 is somewhat useless. Thus, I would suggest dividing income by 10'000 as this is usually done. Age and time might be less problematic, but I can see that it would help – especially for the age interactions to report it in decades rather than years. This would have the additional benefit that the authors could then just report two decimal digits rather than three in their tables. That would tremendously clean-up the tables and make them easier to navigate.*

Thank you very much for this suggestion. We have now rescaled the income variable by 1,000. To avoid confusion as to whether we used continuous, chronological age vs. age group as predictor and as age in years might be easier to interpret than age in decades, we refrain from rescaling the age variable. For reporting precise estimates in the tables, we will continue to present three decimal digits (please note that according to the APA guidelines, statistics such a p scores should be reported with 3 decimal digits).

4. *Tables. I think there are too many tables. This is somewhat overkill. If there is a way to put tables 4 to 7 maybe in the appendix / online supplement, I think that would help. Or if there is a way to reduce these tables in a more compact or condensed form, that would help.*

Thank you for this suggestion. We have now moved Tables 4-7 into the Appendix.

5. *Figures. The figures seem to be all over the place. They have different styles, different fonts, different font sizes, different line widths, etc. Figure 2 has resolution issues, but I am also not sure that figure is actually needed. I don't think people really get that figure and I would rather remove it – or place it in the online supplement. I am even not sure about Figure 1 as the additional benefit over the table details is somewhat meek. Anyway, I would strongly suggest to redo the figures in a coherent and readable style (some of the font sizes are probably too small).*

Thank you for alerting us to this issue. We have now moved Figures 1 and 2 to the Appendix. Additionally, we have solved resolution problems, enhanced the font size of the labels, and now present the figures in a consistent style.

6. *Conclusions. I would like to suggest to include a stronger discussion for the implications and conclusions of the findings. For example, what is the implication for SST or other concepts in the literature of the findings.*

Thank you very much for encouraging us to outline the implications of our results for theories of time perspective. We now discuss (p. 19) that our results warrant a multi-dimensional approach to time perspective (especially concerning SST) but also for SWB. Incorporating a multi-dimensional approach will help to better capture the complex nature of developmental changes in time perspective and well-being. Given the circumscribed evidence for age-related moderations in our study, we propose that this calls for specifying when the proposed age-related benefits of changes in time perspective are present/absent. This could be done by specifying boundary conditions under which age-related changes in time perspective are not related to higher SWB (e.g. when approaching the end of life, Gabrian et al., 2017).

Comments Reviewer 2:

1. *The authors have done an apt and satisfactory job of addressing the key concerns of reviewers (several of which were common across reviewers). In particular, a better case has been made for why the present construct reflects “time perspective” (in light of previous research and theory), and how the work aligns with well-being given that only one measure of life satisfaction was used. Additional information has been provided about the measures and sample as requested, and it has been made more clear how chronological age was used in the analysis (i.e., as a continuous variable, not reflecting distinct age groups) and confusions regarding the reporting of “age group” effects were remedied. Also, some attention has now been given the multidimensional nature of the life satisfaction variable. The additional information added to the Implications sections helped to attenuate my concerns about what these results add to the broader literature. In summary, the present study addresses a standing question in an informative way with the use of multiple time perspective measures and age groups. Contrary to prior research and theoretical predications, limited relationships were observed between time perspectives and life satisfaction. As such, the present findings offer new information about associations between time perspective and life satisfaction during adulthood.*

We thank the reviewer for this positive appraisal of our revised manuscript.